# Medications for opioid use disorder among pregnant women referred by criminal justice agencies before and after Medicaid expansion: A retrospective study of admissions to treatment centers in the United States

Tyler N. A. Winkelman[1,2]*, Becky R. Ford[2], Rebecca J. Shlafer[3], Anna McWilliams[2], Lindsay K. Admon[4], Stephen W. Patrick[5]

**1** Division of General Internal Medicine, Department of Medicine, Hennepin Healthcare, Minneapolis, Minnesota, United States of America, **2** Health, Homelessness, and Criminal Justice Lab, Hennepin Healthcare Research Institute, Minneapolis, Minnesota, United States of America, **3** Department of Pediatrics, University of Minnesota Medical School, Minneapolis, Minnesota, United States of America, **4** Department of Obstetrics and Gynecology, University of Michigan, Ann Arbor, Michigan, United States of America, **5** Vanderbilt Center for Child Health Policy, Departments of Pediatrics and Health Policy, Vanderbilt University School of Medicine, Nashville, Tennessee, United States of America

* tyler.winkelman@hcmed.org

## Abstract

### Background

Criminal justice involvement is common among pregnant women with opioid use disorder (OUD). Medications for OUD improve pregnancy-related outcomes, but trends in treatment data among justice-involved pregnant women are limited. We sought to examine trends in medications for OUD among pregnant women referred to treatment by criminal justice agencies and other sources before and after the Affordable Care Act's Medicaid expansion.

### Methods and findings

We conducted a serial, cross-sectional analysis using 1992–2017 data from pregnant women admitted to treatment facilities for OUD using a national survey of substance use treatment facilities in the United States ($N = 131,838$). We used multiple logistic regression and difference-in-differences methods to assess trends in medications for OUD by referral source. Women in the sample were predominantly aged 18–29 (63.3%), white non-Hispanic, high school graduates, and not employed. Over the study period, 26.3% (95% CI 25.7–27.0) of pregnant women referred by criminal justice agencies received medications for OUD, which was significantly less than those with individual referrals (adjusted rate ratio [ARR] 0.45, 95% CI 0.43–0.46; $P < 0.001$) or those referred from other sources (ARR 0.51, 95% CI 0.50–0.53; $P < 0.001$). Among pregnant women referred by criminal justice agencies, receipt of medications for OUD increased significantly more in states that expanded

**Data Availability Statement:** The data underlying the results presented in the study are available from the Substance Abuse & Mental Health Data Archive at: https://www.datafiles.samhsa.gov/study-series/treatment-episode-data-set-admissions-teds-nid13518.

**Funding:** The authors received no specific funding for this work.

**Competing interests:** The authors have declared that no competing interests exist.

**Abbreviations:** ACA, Affordable Care Act; ARR, adjusted rate ratio; DID, difference-in-differences; OUD, opioid use disorder; pp, percentage point; STROBE, Strengthening the Reporting of Observational Studies in Epidemiology; TEDS-A, Treatment Episode Data Set-Admissions.

Medicaid ($n = 32$) compared with nonexpansion states ($n = 18$) (adjusted difference-in-differences: 12.0 percentage points, 95% CI 1.0–23.0; $P = 0.03$). Limitations of this study include encounters that are at treatment centers only and that do not encompass buprenorphine prescribed in ambulatory care settings, prisons, or jails.

## Conclusions

Pregnant women with OUD referred by criminal justice agencies received evidence-based treatment at lower rates than women referred through other sources. Improving access to medications for OUD for pregnant women referred by criminal justice agencies could provide public health benefits to mothers, infants, and communities. Medicaid expansion is a potential mechanism for expanding access to evidence-based treatment for pregnant women in the US.

## Author summary

### Why was this study done?

- There has been a 4-fold increase in the number of pregnant women with opioid use disorder (OUD). Medications such as methadone and buprenorphine are standard of care for OUD and are recommended during pregnancy, but only 50% of pregnant women receive such medication.

- Pregnant women with OUD who are involved in the criminal justice system are at high risk of poor outcomes, but data regarding the use of medications for OUD in this population are limited.

### What did the researchers find?

- From 1992 to 2017, pregnant women in the US who were referred to treatment for OUD by a criminal justice agency (versus other referral sources) were half as likely to receive medication as part of their treatment plan.

- After implementation of the Affordable Care Act's Medicaid expansion, medication for OUD increased significantly more among pregnant women referred to treatment by criminal justice agencies in Medicaid expansion states compared with nonexpansion states.

### What do these findings mean?

- Pregnant women referred to treatment for OUD by criminal justice agencies were consistently less likely to receive evidence-based treatment, which increases their risk of overdose and poor maternal and neonatal outcomes.

- Improving access to Medicaid for justice-involved individuals may increase the rate at which pregnant women receive evidence-based treatment for OUD.

## Introduction

Overdose deaths among women increased 260% between 1999 and 2017, which was largely driven by a dramatic increase in deaths related to fentanyl, heroin, and prescription opioids [1]. At the same time, the prevalence of opioid use disorder (OUD) among pregnant women more than quadrupled [2,3]. Pregnant women with OUD are more likely to experience severe maternal morbidity and mortality relative to women without OUD at the time of delivery [2]. Treatment of OUD with medication has consistently been shown to improve pregnancy outcomes and reduce the risk of overdose-related death [4–6] compared with non-medication-based treatment, which would typically include individual/group counseling services, withdrawal management, or referrals to other social services. Although effective treatment is available, only half of pregnant women with OUD who enter treatment ultimately receive medication for OUD [7].

Criminal justice involvement (e.g., interaction with police, courts, community supervision, jail, or prison) is common among pregnant women with OUD, with nearly 1 in 5 affected women referred to treatment through criminal justice agencies in 2012 [8]. Since 1980, the incarceration rate among women has increased at a rate two times larger than among men [9]. Criminal justice involvement confers additional risk during pregnancy [10,11] and is associated with significant morbidity and mortality among individuals with OUD [12,13]. Although pregnant women with OUD are increasingly referred to treatment by criminal justice agencies, such as court-mandated treatment or treatment as a condition of probation/parole, there are limited data on trends in medications for OUD for this population [7,8]. Treatment data are critical for leaders in healthcare and correctional settings who have an opportunity to reduce morbidity and mortality among mothers and their infants. Furthermore, since passage of the Affordable Care Act (ACA), insurance rates among individuals involved in the criminal justice system and access to medications for OUD have increased [14–16]. Although the ACA did not alter enrollment criteria for pregnant women, improved Medicaid enrollment strategies, new linkages between community-based healthcare systems and the criminal justice system, and mandated coverage of substance use treatment in states that chose to expand their Medicaid programs could impact treatment patterns for justice-involved pregnant women [17–19].

To assess trends in receipt of medications for OUD among pregnant women with OUD, we examined 25 years of national data from substance use treatment facilities. We compared pregnant women referred to treatment facilities by criminal justice agencies to those with individual referrals (e.g., self-referral to treatment, referred by a friend or family member) and those referred from other sources (e.g., referral from a healthcare provider or employer). We also estimated treatment rates with medications for OUD among pregnant women referred to treatment facilities by criminal justice agencies by state Medicaid expansion status. We hypothesized that pregnant women referred by criminal justice agencies would have lower rates of medications for OUD and that, among this population, Medicaid expansion would be associated with larger increases in medications for OUD relative to nonexpansion states.

## Methods

### Data source and sample

We used all available years (1992–2017) of the Treatment Episode Data Set-Admissions (TEDS-A), an annual national survey of substance treatment facility admissions conducted by the Substance Abuse and Mental Health Services Administration [20]. TEDS-A includes pregnancy status and demographic, substance use, and treatment data that publicly funded treatment facilities are required to report. Data elements are obtained from information included

in the referral itself (e.g., information provided by a criminal justice agency or healthcare provider at the time of referral) and during an interview that is conducted with the patient at the time of intake to the treatment facility. The unit of analysis within TEDS-A is an admission to a treatment facility, not an individual. Therefore, some individuals may be represented more than once. TEDS-A does not contain identifiable information, so it is not possible to link admissions for the same individual. Furthermore, TEDS-A contains information from admissions to treatment centers that accept public funding, and it does not include information about treatment that occurs in other facility types (e.g., primary care offices, in health clinics inside jails and prisons, treatment centers that do not accept public funding). Initial admissions to each center are included; transfers between facilities are excluded.

We restricted the sample to women who were pregnant at the time of admission to treatment and whose primary reason for treatment was related to opioid abuse (heroin, nonprescription methadone, other synthetic opioids). Substance Abuse and Mental Health Services Administration staff corresponded that reporting changes in Florida between 2010 and 2017 made it difficult to compare data across years, so we excluded Florida from the sample. Less than 5% of admissions were missing a variable used in this study. Admissions with missing referral source were excluded from all analyses (2.7%), whereas admissions with missing covariates were excluded from regression models.

### Key independent variable—Treatment referral sources

We examined longitudinal trends in receipt of medications for OUD among pregnant women by referral source to a substance use treatment facility. A referral source is defined as the agency or person referring an individual to treatment. The referral process may vary by jurisdiction. We classified referrals into three categories: criminal justice, individual, and other. Criminal justice agency referrals included referrals to treatment centers from police, probation officers, judges, prosecutors, DUI/DWI court, and parole boards. Criminal justice referrals do not include treatment received within a correctional facility. Individual referrals included admissions that were initiated by the patient, their family, a friend, or an individual who was not included in another category. Other referrals included those from an alcohol or drug abuse care provider, another healthcare provider, schools, employers, or other community referrals (e.g., Alcoholics Anonymous, shelter, or religious organization).

### Key dependent variable—Medications for OUD

Receipt of medications for OUD, such as methadone or buprenorphine, as part of a treatment plan was the primary outcome measure. TEDS-A data do not distinguish between medications. Opioid-related treatment episodes that do not include medications for OUD typically involve individual, family, or group services; withdrawal management; and/or transitional housing. Referrals to treatment do not necessarily ensure an individual will receive medication for OUD and could, instead, result in counseling or other behavioral treatment only.

### Medicaid expansion

Beginning in 2014, states could choose whether to expand their Medicaid programs under the ACA. For purposes of this analysis, we categorized a state as "expanded" in a given year if they had been expanded for more than 6 months that year. We considered most expansion states to have expanded in 2014, with the exception of New Hampshire (2015), Alaska (2016), Montana (2016), and Louisiana (2016).

## Sociodemographic characteristics and service setting

We examined differences between pregnant women with OUD by referral source. We assessed age, race/ethnicity, educational attainment, employment, census region, and service setting. Service setting indicates the location at which an individual received treatment and could be classified as a detoxification, residential, or ambulatory center. We controlled for these characteristics in multivariable models to assess trends in treatment by year.

## Statistical analysis

We prespecified our analysis plan to examine trends in medications for OUD over time among pregnant women who were referred by criminal justice agencies or other sources (S1 Text). After these initial analyses, we conducted an additional difference-in-differences (DID) analysis using previously described specifications [21] to investigate the extent to which the increase in medications for OUD after 2014 was associated with Medicaid expansion and hypothesized that this increase would largely be explained by Medicaid expansion. We first assessed sociodemographic characteristics and service settings of our study population by referral source (i.e., criminal justice, individual, or other referral) between 1992 and 2017. Next, we examined the number of pregnant women with OUD referred by criminal justice agencies or other sources who did and did not receive medications for OUD during our study period. We then used multivariable logistic regression models to estimate the proportion of women who received medications for OUD by referral source overall and in each study year, adjusting for the covariates described above. We used postestimation predictive margins to depict and compare adjusted proportions between referral sources and years. Comparisons are presented as adjusted rate ratios (ARRs).

Finally, we examined changes in receipt of medications for OUD among pregnant women referred by criminal justice agencies by Medicaid expansion status between 2011 and 2017. We first described unadjusted trends by expansion status. Next, we used a DID framework to compare changes in medications for OUD receipt in states that did and did not expand Medicaid. We interacted a state-specific time variable that indicated whether the admission was before or after implementation of the ACA's Medicaid expansion with a variable that indicated whether the admission was in a state that expanded Medicaid during the study period. We used a linear model with robust standard errors clustered at the state level and adjusted for covariates. We did not control for census region in our DID model because we clustered our standard errors at the state level. We excluded 2014 from our adjusted DID model to allow for a washout period [22,23]. We used a linear model with the same covariates to estimate unadjusted and adjusted 2017 rate differences in medications for OUD receipt among pregnant women in states that did and did not expand Medicaid. The primary assumption of a DID design is that trends in comparison groups are parallel prior to the intervention (i.e., Medicaid expansion). Therefore, we assessed trends in medications for OUD by expansion status in the pre-Medicaid expansion time period (2011–2013) by interacting a pre-ACA linear time trend with expansion status in our multivariable linear regression model. Per reviewer request, we also compared our DID estimate among pregnant women referred by criminal justice agencies to women referred by other sources.

We conducted analyses between September 2019 and October 2019. This study is reported as per the Strengthening the Reporting of Observational Studies in Epidemiology (STROBE) guideline (S1 STROBE Checklist). We used Stata 15.1 for all analyses and considered $P < 0.05$ to be statistically significant. This study of publicly available, deidentified data was not considered human subjects research and was exempt from review by the Hennepin Healthcare Research Institute Institutional Review Board.

## Results

From 1992 to 2017, we identified a total of 131,838 pregnant women with OUD; 17,563 (13.3%) were referred by criminal justice agencies, 64,246 (48.7%) were individual referrals, and 50,029 (38.0%) were referred by other sources. Women referred by criminal justice agencies were younger than those referred by an individual or by another source (ages 18–24: 34.1% versus 27.2% versus 29.3%, respectively), were less likely to be Black, non-Hispanic (8.8% versus 16.7% versus 16.2%, respectively), and were more likely to receive treatment in a residential setting (26.2% versus 9.8% versus 20.9%). Women referred by different sources did not differ substantially in education, employment status, or census region (Table 1). During the 25-year study period, more pregnant women referred to treatment for OUD received medications for OUD ($N$ = 67,937; 51.6%) than did not ($N$ = 63,623; 48.4%).

### Medications for OUD by treatment referral source

The number of pregnant women with OUD referred for treatment increased substantially between 1992 and 2017. Pregnant women with OUD referred by criminal justice agencies increased from 318 in 1992 to 1,491 in 2017. Among pregnant women with OUD referred by criminal justice agencies, treatment without medications for OUD was more common than treatment with medications for OUD. Pregnant women with OUD referred by other sources also increased, from 2,325 in 1992 to 9,402 in 2017. In contrast to women referred by criminal justice agencies, pregnant women with OUD referred from other sources were more likely to receive medications for OUD than not (Fig 1).

Adjusted receipt of medications for OUD was significantly lower across the entire study period (1992–2017) for women referred by criminal justice agencies (26.3%, 95% CI 25.7–27.0, $P$ < 0.001) compared with other sources (51.3%, 95% CI 50.8–51.7, $P$ < 0.001; ARR 0.51, 95% CI 0.50–0.53) and individual referrals (59.1%, 95% CI 58.8–59.5, $P$ < 0.001; ARR 0.45, 95% CI: 0.43, 0.46). Longitudinal trends in receipt of medications for OUD among pregnant women also differed by treatment referral source. For example, between 1992 and 2005, the proportion of pregnant women receiving medications for OUD continuously declined over time for those who were referred by criminal justice agencies, whereas similar declines were not seen among other referral sources (Fig 2). Rates of medications for OUD among pregnant women referred by criminal justice agencies remained lower throughout the study period compared with women referred by individual and other sources and never returned to levels from the early to mid-1990s. Adjusted rates of medications for OUD by referral source for each study year are available in the S1 Table of the online supplement.

Pregnant women referred by criminal justice agencies were less likely to receive medications for OUD in 2007 (ARR 0.72, 95% CI 0.57–0.88, $P$ < 0.001) and equally likely in 2017 (ARR 0.98, 95% CI 0.81–1.15, $P$ = 0.80) compared with 1997. Disparities in receipt of medications for OUD between women referred by criminal justice agencies and those referred by other sources changed over time. In 1997, pregnant women referred by criminal justice agencies were less likely to receive medications for OUD than women who were individual referrals (ARR 0.69, 95% CI 0.58–0.79, $P$ < 0.001) or referred from another source (ARR 0.73, 95% CI 0.62–0.85, $P$ < 0.001). This disparity grew by 2007 (criminal justice versus individual referral: ARR 0.44, 95% CI 0.37–0.50, $P$ < 0.001; criminal justice versus other referral: ARR 0.47, 95% CI 0.40–0.54, $P$ < 0.001) and then began to decrease by 2017 (criminal justice versus individual referral: ARR 0.52, 95% CI 0.48–0.55, $P$ < 0.001; criminal justice versus other referral: ARR 0.59, 95% CI 0.55–0.64, $P$ < 0.001). ARRs are available in 5-year increments in the S2 Table of the online supplement.

**Table 1. Demographic characteristics of pregnant women with opioid use disorder by referral source, US, 1992–2017.**

| Characteristic | N (%) | | | P value |
| --- | --- | --- | --- | --- |
| | Criminal Justice (N = 17,563) | Individual (N = 64,246) | Other (N = 50,029) | |
| **Age** | | | | <0.001 |
| 12–17 | 133 (0.8) | 169 (0.3) | 219 (0.4) | |
| 18–24 | 5,996 (34.1) | 17,491 (27.2) | 14,669 (29.3) | |
| 25–29 | 6,027 (34.3) | 21,182 (33.0) | 17,106 (34.2) | |
| 30–34 | 3,363 (19.2) | 14,772 (23.0) | 11,017 (22.1) | |
| 35+ | 2,044 (11.6) | 10,632 (16.6) | 6,964 (13.9) | |
| **Race/Ethnicity** | | | | <0.001 |
| White, non-Hispanic | 12,704 (72.8) | 41,7301 (65.3) | 33,199 (66.8) | |
| Black, non-Hispanic | 1,540 (8.8) | 10,674 (16.7) | 8,070 (16.2) | |
| Hispanic | 832 (4.8) | 3,206 (5.0) | 2,175 (4.4) | |
| Native American/Alaskan Native | 539 (3.1) | 1,339 (2.1) | 1,531 (3.1) | |
| Other | 1,825 (10.5) | 6,923 (10.8) | 4,739 (9.5) | |
| **Education** | | | | <0.001 |
| Less than high school | 6,599 (38.3) | 23,588 (37.3) | 18,870 (38.4) | |
| High school complete | 7,181 (41.7) | 26,549 (42.0) | 19,962 (40.6) | |
| Some college or more | 3,462 (20.1) | 13,115 (20.7) | 10,371 (21.1) | |
| **Employment** | | | | <0.001 |
| Not employed | 15,480 (89.3) | 56,173 (88.6) | 44,554 (90.3) | |
| Employed | 1,853 (10.7) | 7,229 (11.4) | 4,770 (9.7) | |
| **Census Region** | | | | <0.001 |
| Northeast | 5,520 (31.4) | 19,665 (30.6) | 18,569 (37.1) | |
| Midwest | 3,391 (19.3) | 11,286 (17.6) | 10,234 (20.5) | |
| South | 3,952 (22.5) | 17,111 (26.6) | 12,211 (24.4) | |
| West | 4,700 (26.8) | 16,184 (25.2) | 9,015 (18.0) | |
| **Service Setting** | | | | <0.001 |
| Detox | 708 (4.0) | 7,947 (12.4) | 3,461 (6.9) | |
| Residential | 4,587 (26.2) | 6,291 (9.8) | 10,444 (20.9) | |
| Ambulatory | 12,229 (69.8) | 49,841 (77.8) | 35,985 (72.1) | |

"Criminal Justice" includes referral from police, probation officers, judges, prosecutors, DUI/DWI court, or parole board. "Individual" includes referral from patient, family, or friends. "Other" includes referral from alcohol/drug abuse care provider, healthcare providers, school, employer, or community referral.

Abbreviations: DUI, driving under the influence; DWI, driving while intoxicated

## Medications for OUD among pregnant women referred by criminal justice agencies by state Medicaid expansion status

We examined changes in medications for OUD receipt among pregnant women with OUD referred by criminal justice agencies in states that did and did not expand Medicaid through the ACA (Fig 3). In Medicaid expansion states, rates of medications for OUD were relatively consistent between 2011 and 2013. Medicaid nonexpansion states had lower rates of medications for OUD among pregnant women referred by criminal justice agencies than expansion states. Unadjusted rates of medications for OUD by Medicaid expansion status are available in the S3 Table of the online supplement.

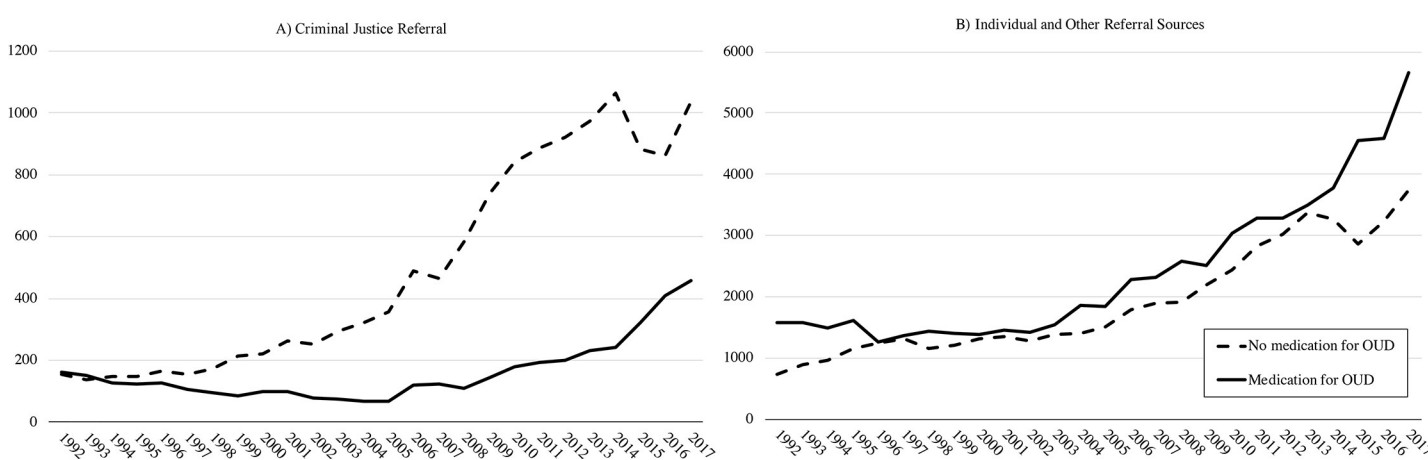

**Fig 1. Number of pregnant women who did and did not receive medications for OUD by referral source, US, 1992–2017.** OUD, opioid use disorder.

Between 2011–2013 and 2015–2017, medications for OUD increased to a greater degree among pregnant women with a criminal justice referral with OUD in states that expanded Medicaid (DID 11.7 percentage points [pp], 95% CI 0.5–22.9; $P = 0.04$), and this difference remained significant after adjusting for covariates (adjusted DID 12.0 pp, 95% CI 1.0–23.0; $P = 0.03$). The DID estimate among pregnant women referred by criminal justice agencies was similar to individual referrals (adjusted DID 11.9 pp, 95% CI 2.3–21.4; $P = 0.02$) and other referral sources (adjusted DID 13.4 pp, 95% CI −1.3 to 28.0; $P = 0.07$). In 2017, pregnant women referred by criminal justice agencies were significantly more likely to have received medications for OUD in states that expanded Medicaid compared with women in states that did not expand Medicaid (unadjusted rate difference: 26.6 pp, 95% CI 10.2–43.0; adjusted rate difference: 27.2 pp, 95% CI 11.3–43.0). Adjusted rates of medications for OUD by Medicaid expansion status are available in the S4 Table of the online supplement. Trends in pre-ACA rates of medications for OUD did not vary significantly by expansion status (interaction coefficient: 0.1%, 95% CI −4.7 to 4.9).

## Discussion

From 1992 to 2017, pregnant women with OUD referred to substance use treatment facilities by criminal justice agencies were consistently less likely to receive medications for OUD than pregnant women referred by other sources. OUD during pregnancy is associated with low birth weight, preterm labor, fetal death, and increased severe maternal morbidity [2,4,24]. Medications for OUD, the recommended first-line treatment for OUD in pregnancy, reduces overdose risk and improves maternal and infant outcomes regardless of criminal justice involvement [5,25]. Persistent disparities in medications for OUD among women referred by criminal justice agencies suggest structural and philosophical barriers to evidence-based treatment that negatively impact the health of mothers involved in criminal justice agencies and their infants [25].

Our finding of low rates of medications for OUD among pregnant women referred by criminal justice agencies is consistent with studies in other justice-involved populations with OUD [12,26,27]. For example, medications for OUD are available to fewer than half of drug court participants [26]. Barriers to medications for OUD in drug courts include cost, lack of provider, concerns about diversion, and stigma [26]. Similar barriers also likely contribute to the disparities in medications for OUD found in our study population. Poor linkage to

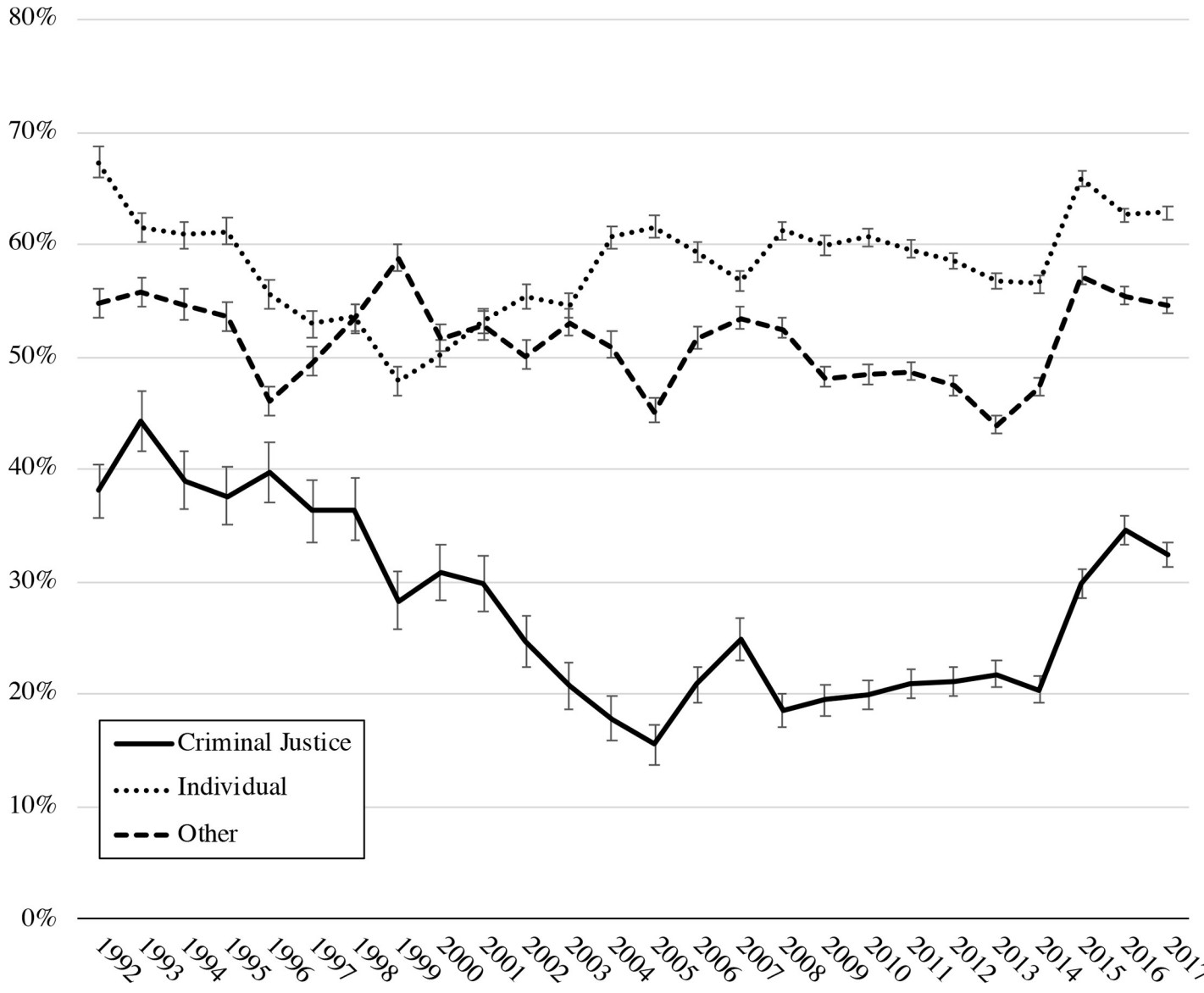

**Fig 2. Adjusted proportion of pregnant women receiving medications for opioid use disorder by referral source, US, 1992–2017.** Adjusted for age, race/ethnicity, educational attainment, employment, census region, and service setting.

evidence-based OUD treatment within the criminal justice system is a critical public health issue because many individuals with OUD interact with the criminal justice system and, subsequently, have a high risk of death from opioid overdose upon release [13].

To address the opioid crisis, researchers have called for the implementation of a cascade of care that effectively links high-risk populations, especially individuals with criminal justice involvement, to OUD treatment [28]. In addition, sequential intercept mapping, a systems-based model to divert individuals with behavioral health issues from the criminal justice system to community-based treatment, is a promising approach to identify points within the criminal justice system at which alternative strategies could improve care for individuals, including pregnant women, with OUD [29]. Policy responses to substance use during pregnancy that incorporate both criminal justice and public health interventions have been associated with higher levels of treatment, whereas approaches focused solely on criminal justice

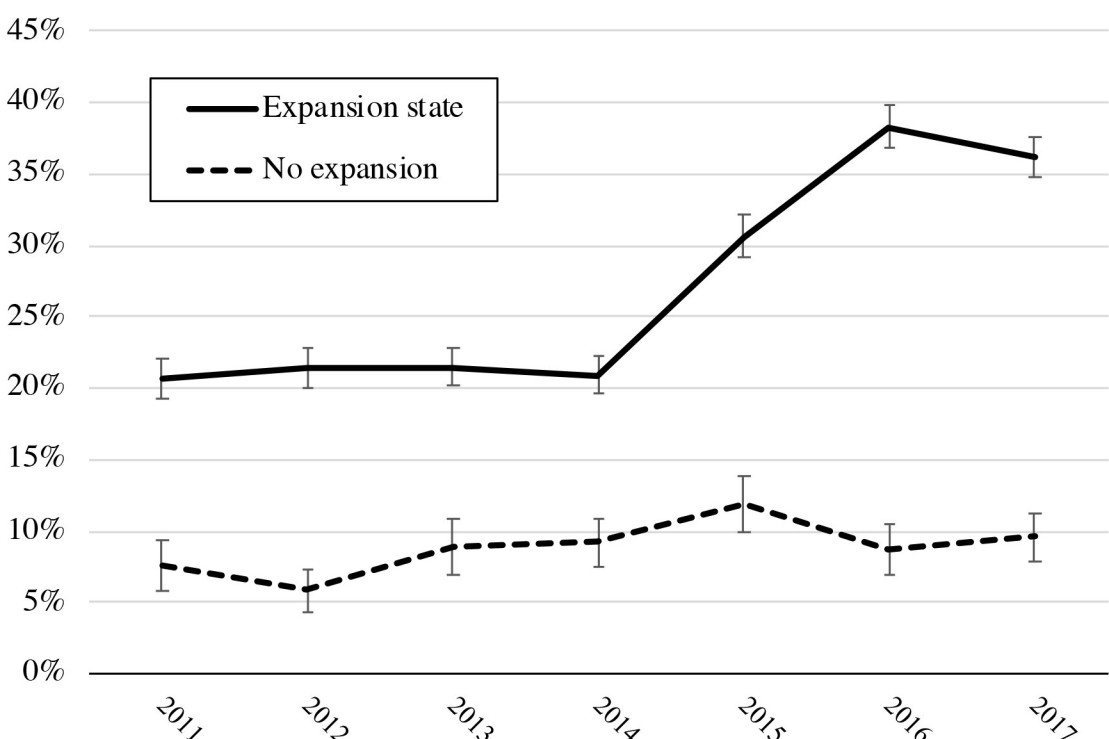

**Fig 3. Proportion of pregnant women referred by criminal justice agencies receiving medications for OUD by state Medicaid expansion status, US, 2011–2017.** OUD, opioid use disorder.

policy have not improved treatment rates [30,31]. Thus, our findings, in conjunction with previous data, indicate that a robust cross-sector approach to opioid use during pregnancy is required to increase medications for OUD among pregnant women with OUD. Improving medications for OUD rates for pregnant women referred by the criminal justice system could provide benefits to both public safety and public health [30,31].

Although the ACA did not alter Medicaid coverage during pregnancy [32], we found that the ACA's Medicaid expansion was associated with a significant increase in medications for OUD among pregnant women with a criminal justice agency or individual referral in expansion states compared with nonexpansion states. There are several possible explanations for this finding. First, the ACA improved coverage for women of reproductive age [33]. Thus, pregnant women may be more likely to start their pregnancy, and subsequently treatment, with insurance. Insurance status may alter referral patterns and increase referrals to centers that provide medications for OUD. Second, beginning in 2014, there were numerous efforts to link individuals to health insurance upon release from correctional facilities [17]. Such linkage could increase insurance levels among recently incarcerated women who qualified for insurance coverage prior to the ACA but were not covered because of difficulties with the enrollment process. Medicaid is often terminated during incarceration, and individuals must reenroll upon release [34]. Low-income women enrolled in Medicaid are likely to have better access to medications for OUD than women who are uninsured [35]. Finally, Medicaid expansion is associated with increased uptake of medications for OUD in the general population, partly because Medicaid expansion was mandated to cover behavioral health services [19]. Therefore, although coverage did not change during pregnancy, the treatments covered by

Medicaid expanded, which could have increased medications for OUD among this population [36].

## Limitations

The findings from this study are not without limitation. First, though TEDS-A is the most comprehensive survey of treatment admissions in the US, some states only report clients whose care was publicly funded, though most states report all eligible admissions [37]. Omission of some privately funded admissions could potentially alter our results if reporting varied substantially between expansion and nonexpansion states. Second, our data set includes medications for OUD only at treatment facilities. The number of providers who prescribe buprenorphine from an outpatient setting increased over 600% between 2006–2008 and 2012–2014, but visits with these providers are not available through TEDS-A [38]. Furthermore, TEDS-A only includes data from treatment centers in the community, and individuals who are receiving treatment inside of a correctional facility (prison or jail) are not included. Third, payer is missing for most admissions, and thus we could not assess changes in payer for treatment admissions. Fourth, TEDS-A only accounts for individuals referred to treatment by criminal justice agencies and thus is a conservative estimate of criminal justice involvement in this population. Many individuals with OUD who are not linked to treatment have involvement with the criminal justice system, and some individuals with criminal justice involvement will self-refer to treatment without an actual referral from a criminal justice agency. Fifth, our data do not allow us to infer why pregnant women referred by criminal justice agencies receive medications for OUD less often than those referred by other sources. For example, it is possible that those referred by criminal justice agencies are referred to different facilities because of state/county contracts with treatment centers or that those with criminal justice involvement refuse medications because of conditions of community supervision or mistrust of the healthcare system. Finally, our DID model can identify associations between policy and treatment utilization but is not an experimental design.

## Conclusion

Pregnant women with OUD referred by criminal justice agencies between 1992 and 2017 were consistently less likely to receive evidence-based treatment than women referred through other sources. The ACA's Medicaid expansion was associated with significant improvements in receipt of medications for OUD among pregnant women referred by criminal justice agencies. Expansion of Medicaid in states that have not yet taken it up would likely further improve OUD treatment among pregnant women. A cross-sector approach that links pregnant women with OUD to care is needed to stem rising rates of opioid-related morbidity and mortality and address persistent disparities between pregnant women with and without involvement in the criminal justice system.

## Supporting information

**S1 Table. Adjusted rates of medications for opioid use disorder by treatment referral source.**
(XLSX)

**S2 Table. Adjusted rate ratios of medications for opioid use disorder in 5-year increments.**
(XLSX)

**S3 Table. Unadjusted rates of medications for opioid use disorder among pregnant women in Medicaid expansion and nonexpansion states for each referral source.**
(XLSX)

**S4 Table. Adjusted rates of medications for opioid use disorder for pregnant women referred to treatment in Medicaid expansion and nonexpansion states for each referral source.**
(XLSX)

**S1 STROBE Checklist. STROBE, Strengthening the Reporting of Observational Studies in Epidemiology.**
(DOCX)

**S1 Text. Opioid use disorder treatment among pregnant women involved in the criminal justice system.**
(DOCX)

## Author Contributions

**Conceptualization:** Tyler N. A. Winkelman, Rebecca J. Shlafer, Lindsay K. Admon, Stephen W. Patrick.

**Formal analysis:** Tyler N. A. Winkelman, Becky R. Ford.

**Investigation:** Tyler N. A. Winkelman.

**Methodology:** Tyler N. A. Winkelman, Becky R. Ford, Rebecca J. Shlafer, Lindsay K. Admon, Stephen W. Patrick.

**Project administration:** Becky R. Ford, Anna McWilliams.

**Supervision:** Tyler N. A. Winkelman.

**Writing – original draft:** Tyler N. A. Winkelman, Becky R. Ford, Anna McWilliams, Lindsay K. Admon, Stephen W. Patrick.

**Writing – review & editing:** Tyler N. A. Winkelman, Becky R. Ford, Rebecca J. Shlafer, Lindsay K. Admon, Stephen W. Patrick.

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
