## [Decision Letter · Decision Letter 0]

3 Feb 2020

Dear Dr. Winkelman,

Thank you very much for submitting your manuscript "Medicaid expansion and receipt of medications for opioid use disorder among pregnant women referred by the criminal justice system" (PMEDICINE-D-19-03648) for consideration at PLOS Medicine. 

[LINK]

In light of these reviews, I am afraid that we will not be able to accept the manuscript for publication in the journal in its current form, but we would like to consider a revised version that addresses the reviewers' and editors' comments. Obviously we cannot make any decision about publication until we have seen the revised manuscript and your response, and we plan to seek re-review by one or more of the reviewers. 

We expect to receive your revised manuscript by Feb 24 2020 11:59PM. Please email us (plosmedicine@plos.org) if you have any questions or concerns.

We look forward to receiving your revised manuscript. 

Sincerely,

Caitlin Moyer, Ph.D.

Associate Editor 

PLOS Medicine

plosmedicine.org

The Academic Editor raised a point and we would ask you to discuss in the manuscript:

The Medicaid result is quite strong. However, the other result about criminal justice referral is less strong, because there are potentially unobserved attributes of people referred by one mechanism or another that are also correlated with their receipt of medications for OUD. It is not possible to control for all of those things, so I worry about the issue of endogeneity (of confounding) there.

Editorial points and queries: 

Please revise your title according to PLOS Medicine's style. Your title must be nondeclarative and not a question. It should begin with main concept if possible. "Effect of" should be used only if causality can be inferred, i.e., for an RCT. Please place the study design ("A randomized controlled trial," "A retrospective study," "A modelling study," etc.) in the subtitle (ie, after a colon). Also please add the country in which the study is set. 

Abstract- Please add summary demographic information of the women included in the study; please add p values (and elsewhere I tables / text) where 95%Cis are given; the final sentence of the Methods and Findings section should include a description of the studies limitations. 

Data – the URL provided returns ‘page not found’. PLOS Medicine requires that the de-identified data underlying the specific results in a published article be made available, without restrictions on access, in a public repository or as Supporting Information at the time of article publication, provided it is legal and ethical to do so. Please see the policy at 

http://journals.plos.org/plosmedicine/s/data-availability

and FAQs at 

http://journals.plos.org/plosmedicine/s/data-availability#loc-faqs-for-data-policy

Please avoid bold font in main text for emphasis

Did your study have a prospective protocol or analysis plan? Please state this (either way) early in the Methods section.

c) In either case, changes in the analysis—including those made in response to peer review comments—should be identified as such in the Methods section of the paper, with rationale.

Screen reader support enabled.

Please ensure that the study is reported according to the [STROBE] guideline, and include the completed [STROBE or other] checklist as Supporting Information. When completing the checklist, please use section and paragraph numbers, rather than page numbers. Please add the following statement, or similar, to the Methods: "This study is reported as per the Strengthening the Reporting of Observational Studies in Epidemiology (STROBE) guideline (S1 Checklist)."

Please report your study according to the relevant guideline, which can be found here: http://www.equator-network.org/

Comments from the reviewers:

Reviewer #1: Statistical review

This paper reports a repeated cross-sectional study that investigates the changes in treatment given to pregnant women with opioid dependency. In particular the authors look at how this changes after Medicaid was expanded in a group of states compared to other states that did not expand it.

I have some comments on the statistical methods and reporting, which I have summarised below:

1. From what I understand there was a big political difference in the two types of states (expansion vs non-expansion). From what I understand from natural experiments, to get robust causal conclusions you'd want the two groups to be comparable at the time the change is made (perhaps after an adjustment through matching). In this case the prescription behaviour seems rather different both before and after the Medicaid expansion. Is there strong potential for confounding in this? Would matching by other characteristics of states be possible? Having said all that, the results do look fairly convincing in terms of the trajectory before and after expansion.

2. Abstract - would be useful to have effect sizes + CIs for the difference between the referral paths rather than just CIs for each arm separately.

3. Abstract - number of expansion and non-expansion states?

4. Line 83 - is it possible to link admissions of the same individual? If so, are there many individuals that are seen multiple times? If so, did the authors consider using a method that would allow for correlation within admissions from the same individual?

5. Line 144/145 - I understand the rationale for the washout period, but did this make a large difference to the results? Would be interesting to include 2014 to see (unless the pre-specified analysis plan excluded 2014).

6. Statistical analysis - was there any missing demographic data, or were individuals only included with complete data? Any issues of representativeness in that case?

7. Line 156 - p-values are mentioned but no p-values are given, so I don't think this sentence is needed.

8. Would be interesting to see whether same results for expansion vs non-expansion would be the case for the other referral paths too - why just looking at prison?

James Wason

Reviewer #2: This is an analysis of 1992-2017 data on 131,838 pregnant women in the US who entered treatment for opioid use.

The authors found that a large proportion of women did not receive standard of care treatment with medications, and furthermore, that those referred from the criminal justice system received even less appropriate treatment with medications. The authors also found that, not surprisingly, receipt of medications was higher in Medicaid expansion states.

The authors did not include Florida- would it be possible to look at the Florida data and see, at least if the data that is available is consistent with the rest of the country? If not, how so? The state is a large one and it would be re-assuring if the data there were at least consistent.

Reviewer #3: Thank you for the opportunity to review this manuscript. This is a novel study on an important and overlooked topic that, at its core, seems to suggest discrimination against pregnant people referred for treatment from criminal justice agencies, that they receive worse treatment than those without. The statistical approach is a compelling one to highlight this problem. It's a promising paper with an innovative approach with a variable that receives little attention when it comes to pregnant people, criminal justice system involvement. This paper would be greatly strengthened by providing more detail and clarity on a number of methodological/definitional fronts to better orient the reader and contextualize the meaning of and limits of the data we can know from this data source; specific suggestions on where to do this are outlined below. 

Title, abstract, and throughout uses the term "the criminal justice system." This makes it sound like a monolithic entity, which it is not, as the authors are well aware. In calling it that, it's confusing to know whether referrals came from jails, prisons, drug courts, probation, parole. This monolithic descriptor makes it seem like they are all the same, when they all have very different processes. This should be clarified and defined up front in the abstract and the intro. It takes getting deep into the methods section to realize that this actually does not include prisons or jails, which is important to explain up front. 

Perhaps saying criminal justice system agencies is better (as in line 57), and then being very explicit in the introduction and discussion that these are community based CJS agencies, or some other way of saying that this does not include jails and prisons.

Abstract-Lines 26-27-could use some more clarity in the first sentence. "data of

pregnant women in the United States who reported opioids as their primary reason for treatment

 (N=131,838)." It's confusing not knowing the source of the data and opioids as primary reason for treatment from what kind of provider or agency. . . While the manuscript will get into more detail of course, this first sentence confused me as to how various data sources were defined. It's also confusing without that context what these women would be referred for if not for M-OUD. Are they just being referred for prenatal care? Or is it drug treatment referrals you are talking about and they just get behavioral treatment? Just mentioning that the data come from a national survey of admissions to substance treatment facilities would be orienting for the reader.

Intro/Lines 60-66: This part of the intro needs more explanation as to what you mean by 'pregnant women referred by the criminal justice system.' Does this mean people who are not in custody at the time of the referral? Which kinds of agencies? How do CJS agency referrals for MOUD even happen? Are they court mandated referrals? The "referral" aspect is the crux of the study, but I think it could use a little more specificity as to how/what this really means. Relatedly, it would also be helpful to contextualize what "other sources" means. It likewise sounds monolithic but likely is not, and would be useful to elaborate more on what you mean by this, in contradistinction to criminal justice agencies. While the methods section gets into this more, the intro needs some of this overall context (though not as much specificity as in the methods)—also because once you read the methods you see that "the criminal justice system" as available in the database and therefore as defined in this study actually does not include jails or prisons. 

Intro would also benefit from some information on trends in women's involvement in criminal justice system agencies that are being described—pregnant women data would be ideal, but I suspect this is not available. This would just be helpful so that readers have some sense of the trends in female CJS involvement over time. 

Methods- 

78-85: Clarify that this is a survey done annually (if indeed it is). Perhaps I missed this mention, but it's worth saying here in the methods that it is an annual survey. 

87-91- Please explain how this survey asks about/records/verifies pregnancy status, so we know how the sample population is ascertained.

94-95: Could use a brief few words that 'referral for treatment' could be just for behavioral treatment—to underscore that medication treatment is something separate that is not necessarily a given with a referral.

118-123: How are these variables assessed by the person reporting? Do we know? Is it just "reported race/ethnicity" etc? Also, how does this survey distinguish for-profit facilities that do not accept insurance? The limitations section mentions something about this, but in explaining the database to the readers in the methods, some of this information is helpful up front.

Discussion

This section too needs a reminder that this is talking about people in the community who are referred by criminal justice system agencies-- that this is not about incarcerated people. 

Somewhere in the discussion it's also worth noting, even as the authors note that the database does not distinguish methadone vs. buprenorphine, that buprenorphine did not become as widely used in pregnancy until after the results of the MOTHER trial were published in 2012. Is there some way that this might come into play? 

The discussion section should acknowledge that there may be other reasons pregnant people are not on M-OUD—including that they decline medication treatment. This study could not assess for people who voluntarily do not want to me on medication. And people referred by CJS agencies are different than those from other sources, eg. may have experienced more coercion, mistrust etc, and this may account for some of the difference (maybe CJS referred people were offered medications but were more likely to decline) which this study cannot measure. 

We cannot assume that people who had referrals by self or other did not have CJS involvement. They certainly might, it's just that a CJS agency wasn't the one who referred them to treatment. So can we really isolate this as CJS effect? Maybe people who were told by a CJS agency that they had to be in treatment didn't want to accept medication treatment offered to them, whereas women who referred themselves, even if they might have also had CJS involvement, would be more likely to be open to medication. So it's not just that as stated in lines 315-316 this study can't account for people with CJS involvement who were not referred, it's also that people who self referred may also have CJS involvement, it's just not mandated/part of the referral.

[LINK]

---

## [Decision Letter · Decision Letter 1]

2 Apr 2020

Dear Dr. Winkelman,

Thank you very much for re-submitting your manuscript "Medications for opioid use disorder among pregnant women referred by criminal justice agencies before and after Medicaid expansion: A retrospective study of admissions to treatment centers in the United States" (PMEDICINE-D-19-03648R1) for review by PLOS Medicine.

I have discussed the paper with my colleagues and the academic editor and it was also seen again by reviewers. I am pleased to say that provided the remaining editorial and production issues are dealt with we are planning to accept the paper for publication in the journal.

[LINK]

We look forward to receiving the revised manuscript by Apr 09 2020 11:59PM. 

Sincerely,

Caitlin Moyer, Ph.D.

Associate Editor 

PLOS Medicine

plosmedicine.org

Requests from Editors:

Abstract – please add summary demographic information, including mean age and also is it possible to say which cities or is it truly nationwide (this would also be helpful in the main text?; Can you please add a sentence of limitations as the final sentence of the ‘Methods and Findings section’, or at least be more explicit by starting that sentence with ‘Limitations of our study are……’

Where p values are for example .2, please write as 0.2

STROBE checklist – Please amend your checklist as it currently does not appear to have sections and paragraphs. The STROBE guideline can be found here: http://www.equator-network.org/reporting-guidelines/strobe/ When completing the checklist, please use section and paragraph numbers, rather than page numbers.

Comments from Reviewers:

Reviewer #1: Thank you to the author for addressing my previous comments well. I have no further issues to raise.

Reviewer #3: The authors have an outstanding job revising this manuscript and responding to my initial review. The article provides important insights and I recommend this article for publication.

I have two extremely minor suggestions:

Line 325- consider adding a citation to this new publication, which describes use of medications for OUD for pregnant and postpartum women in jails and prisons.

https://www.ncbi.nlm.nih.gov/pubmed/32141128

Line 353- add "in" between early and pregnancy

Thank you for the opportunity to review this revised manuscript.

[LINK]

---

## [Editor Report · Decision Letter 2]

22 Apr 2020

Dear Dr. Winkelman, 

On behalf of my colleagues and the academic editor, Dr. Zirui Song, I am delighted to inform you that your manuscript entitled "Medications for opioid use disorder among pregnant women referred by criminal justice agencies before and after Medicaid expansion: A retrospective study of admissions to treatment centers in the United States" (PMEDICINE-D-19-03648R2) has been accepted for publication in PLOS Medicine. 

PRODUCTION PROCESS

PRESS

PROFILE INFORMATION

Thank you again for submitting the manuscript to PLOS Medicine. We look forward to publishing it. 

Best wishes, 

Caitlin Moyer, Ph.D.

Associate Editor 

PLOS Medicine

plosmedicine.org